# Cloud-native simulation framework for gossip protocol: Modeling and analyzing network dynamics

**Samsuddin Samsuddin Wira**[1]*, **Chee Keong Tan**[1],
**Wai Peng Wong**[1], **Ian K.T. Tan**[2]

**1** School of Information Technology, Monash University Malaysia, Sunway, Selangor, Malaysia, **2** School of Mathematical and Computer Sciences, Heriot-Watt University Malaysia, Putrajaya, Selangor, Malaysia

☯ These authors contributed equally to this work.
* samsuddin.samsuddin@monash.edu

**Data availability statement:** Data available at our GitHub repository: https://github.com/wwiras/cnsim

## Abstract

This research paper explores the implementation of gossip protocols in cloud native framework through network modeling and simulation analysis. Gossip protocol is known for their decentralized and fault-tolerant nature. Simulating gossip protocols with conventional tools may face limitations in flexibility and scalability, complicating analysis, especially for larger or more diverse networks. In this paper, gossip protocols are tested within the context of cloud native computing, which leverages its scalability, flexibility, and observability. The study aims to assess the performance and feasibility of gossip protocols within cloud-native settings through a simulated environment. The paper delves into the theoretical foundation of gossip protocol, highlights the core components of cloud native computing, and explains the methodology employed in the simulation. A detailed guide has been provided on utilizing cloud-native frameworks to simulate gossip protocols across varied network environments. The simulation analysis provides insights into gossip protocols' behavior in distributed cloud-native systems, evaluating aspects of scalability, reliability, and observability. This investigation contributes to understanding the practical implications and potential applications of gossip protocol within modern cloud-native architectures, which can also apply to conventional network infrastructure.

## Introduction

Gossip protocol, in simple terms, resembles the way rumors spread in a social setting [1]. It is a method used by computer systems to communicate with each other, much like people sharing news. In this system, a computer shares information with a few others. Those, in turn, pass this information to more computers. This process continues, ensuring that all computers eventually get the update.

The beauty of this approach comes from its simplicity and efficiency [2,3]. Unlike traditional methods where a central server sends updates to every computer, gossip protocol reduces the load on any single machine. This method is beneficial in large networks, where direct communication between all computers is impractical. It ensures quick and widespread

**Funding:** Internal Matching Grant - Public Service Department (JPA) of Malaysia. The funders had no role in study design, data collection and analysis, decision to publish, or preparation of the manuscript.

dissemination of information, making it a preferred choice in various applications, from updating data across servers to spreading news in social networks.

Fig 1 demonstrates the operation of the gossip protocol within a network consisting of multiple nodes. The network is inactive until Node A initiates the gossip protocol by sending a message to its connected neighboring nodes (i.e., Node B, Node C and Node D). This activity triggers the flow of information through the network, ensuring that every node receives the message within the next iteration. It highlights the decentralized nature of gossip protocols, where each node contributes independently to the dissemination of information, ensuring resilience and eliminating the need for a centralized coordinating entity. The essence of

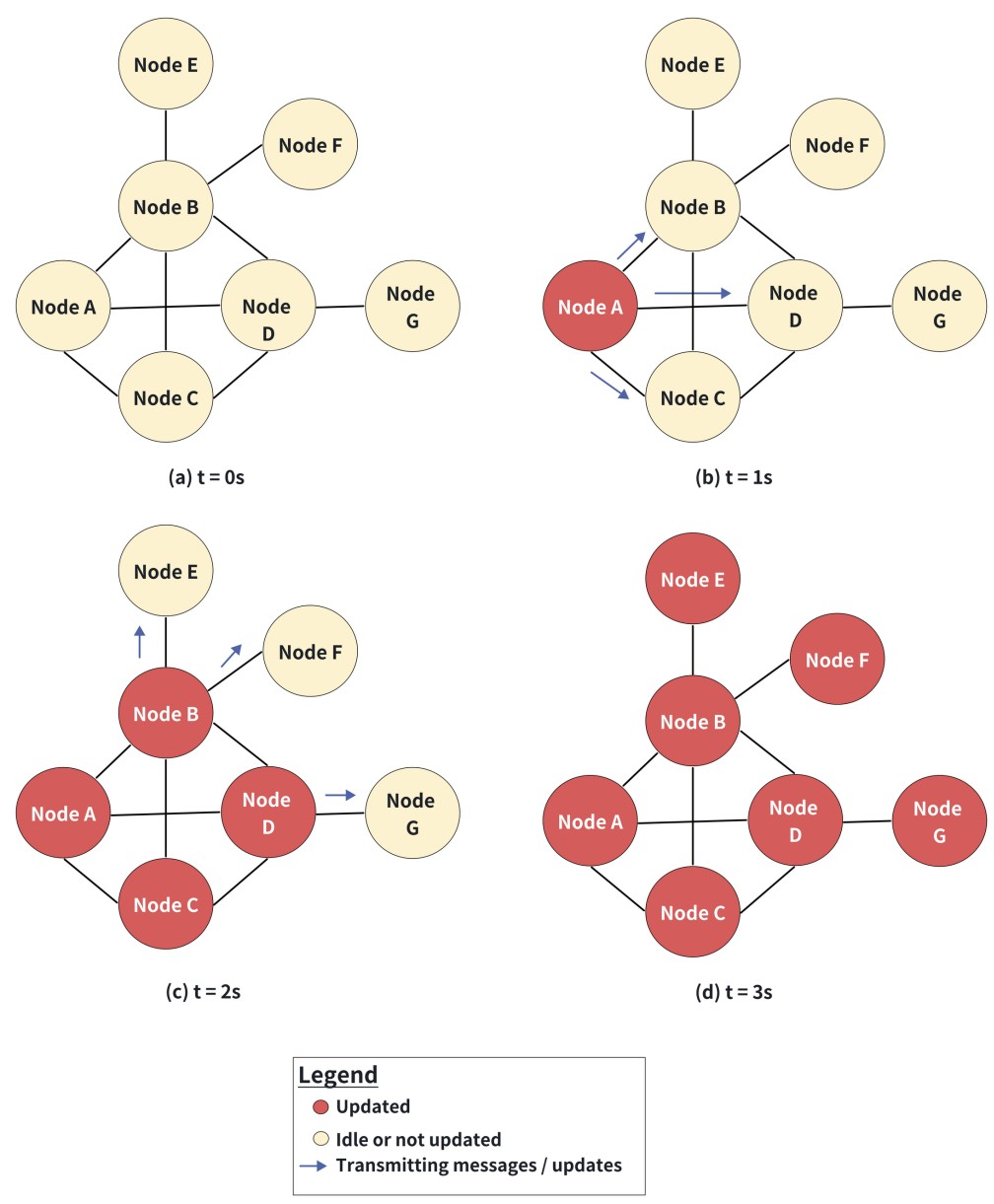

**Fig 1. Gossip protocol dissemination illustration.**

gossip protocols in achieving rapid and reliable communication within distributed systems is succinctly captured through this process.

Among various applications, the blockchain network stands out as a prominent example showcasing the implementation of the gossip protocol. In such a network, it is crucial to maintain the integrity and consistency of the ledger across multiple nodes. Gossip protocols in blockchain help enhance data distribution and make the network resilient against failures and malicious attacks, ensuring its robustness and reliability. With the increasing adoption of blockchain technology across various industries [4,5] the significance of gossip protocols has become more pronounced. These protocols can support scalable, secure, and efficient communication within large, distributed networks, underscoring their critical role in the foundational technology of the digital age. The widespread use of gossip protocols in blockchain networks exemplifies their value and indispensability in modern distributed computing environments.

Gossip protocols find diverse applications beyond blockchain networks, demonstrating versatility in distributed systems, peer-to-peer networks, and cloud computing environments. In distributed systems, such as content delivery networks (CDNs) and distributed databases, gossip protocols facilitate efficient dissemination of information, ensuring data consistency and fault tolerance. In peer-to-peer networks, protocols like BitTorrent utilize gossip-based mechanisms for file sharing, enabling decentralized and scalable distribution of large files. Additionally, in cloud computing environments, gossip protocols play a crucial role in service discovery, load balancing, and cluster management, enhancing scalability and resilience. These applications underscore the broad utility of gossip protocols in various distributed and decentralized systems beyond the realm of blockchain technology.

On the contrary, cloud-native computing signifies a paradigm shift in the development and deployment of applications, harnessing the transformative power of cloud technology to its fullest extent. This approach not only emphasizes containerization and microservices architecture but also advocates for agile development practices and continuous integration/continuous deployment (CI/CD) pipelines. By enabling scalability, flexibility, and resilience, cloud-native computing empowers organizations to innovate rapidly and efficiently in today's dynamic digital landscape [6] . Unlike traditional computing, where applications are often designed for a specific set of hardware, cloud-native applications are built from the ground up to thrive in a dynamic, virtualized cloud environment. This approach offers significant advantages such as resilience, efficiency, and flexibility [7].

Critical components of cloud-native computing include software containers [8], microservices [9], software-defined infrastructure [10], and application program interfaces (APIs) [11]. Containers allow applications to be packaged with all their dependencies, facilitating portability across different cloud environments. Microservices enable the construction of applications as a collection of small, independent services, enhancing flexibility and scalability. Software-defined infrastructure virtualizes hardware resources, allowing for more agile and cost-effective scaling. APIs, on the other hand, provide the means to connect and extend applications with minimal coding.

Cloud-native applications are inherently adaptable, scalable, and portable [12]. They can efficiently respond to business needs, scaling resources up or down as required. Their portability ensures they can run on various platforms, from smartphones to mainframes. In contrast, legacy systems run in the cloud but only partially exploit these benefits. Legacy applications often need more scalability, extensibility, and efficiency due to their reliance on server-centric architectures.

The exploration network modeling and simulation for gossip protocols within a cloud-native framework presents a compelling narrative of advancement over traditional simulation

methodologies. By closely examining the limitations of current simulators, the adoption of cloud-native principles in gossip protocol simulations emerges as a superior approach. The following are vital benefits underpinning this potential:-

- **Simplicity and Efficiency vs. Limited Scalability and Flexibility**

  Traditional simulators, often confined within a legacy system architecture [3,13–16], typically restrict simulations to a mere handful of nodes despite possessing high-end server capabilities. This constraint contrasts with the distributed communication model facilitated by gossip protocols in cloud-native environments, where loads are effectively spread across numerous nodes to enhance performance and directly tackle traditional simulations' scalability and flexibility issues.

- **Resilience vs. Dependency on Monolithic Architectures**

  Legacy simulations, especially those reliant on MATLAB [17], incorporate monolithic architectures that limit their ability to harness the full potential of cloud-native environments, notably in resilience aspects. However, cloud-native application designs ensure reliable dissemination of information across the network, even in cases of individual component failures, effectively overcoming the resilience issues associated with monolithic architectures.

- **Scalability vs. Inefficient Resource Utilization**

  Simulators like PeerSim [18,19], capable of handling environments with millions of nodes, find their scalability and flexibility could be improved by legacy system architectures, leading to efficient resource utilization. On the other hand, cloud-native computing facilitates dynamic resource allocation, allowing for efficient management of networks of various sizes and complexities, thereby addressing the scalability challenges such simulators encounter.

- **Flexibility and Portability vs. Limited Adaptability**

  The use of legacy infrastructures, such as Virtual Machines (VMs) [3,13–16], is marked by slower performance, less efficient resource utilization, and reduced scalability and portability. Moreover, VMs face challenges in adaptability. Cloud-native environments provide superior flexibility and portability, enabling simulations to rapidly adapt to changes and run seamlessly across different platforms, thus resolving the adaptability issues observed in VM-based simulations.

- **Cost-Effectiveness vs. High Operational Expenses**

  Traditional computing models [3,13–16], reliant on fixed infrastructure and inefficient resource management, incur high operational expenses. Contrastingly, cloud-native setups strategically manage computing resources, enabling more extensive and complex simulations without significantly increasing costs, presenting an economically efficient solution for advanced simulation tasks.

- **Observability vs. Lack of Detailed Performance Insights**

  A notable area for improvement in existing simulators is their overlook of the potential benefits of cloud-native observability features, leading to a dearth of detailed performance insights. This oversight restricts the optimization of simulations based on real-time data. Cloud-native systems, equipped with advanced monitoring and observability tools, offer comprehensive insights into performance metrics in real-time, facilitating precise optimization and effectively addressing the observability shortcomings in previous simulations [20,21].

The combination of gossip protocol with cloud native computing has yet to be fully explored in academic research [22]. Interestingly, from our extensive review and analysis, no

research has extensively explored this specific intersection of gossip protocol simulation and cloud-native environments, highlighting this study's novelty and potential impact. To address this, our paper explores the potential applications of implementing gossip protocol within cloud-native settings. This project is essential for several reasons, particularly as organizations increasingly gravitate towards cloud-native architectures and strive for efficient implementation of effective communication techniques amidst containerized microservices. The use of gossip protocols presents a valuable solution to meet these flexible requirements.

Moreover, even though gossip protocol have exhibited durability [23], there is still a lack of exploration regarding their implementation and performance in cloud-native environments. By simulating this protocol under such conditions, both advantages and disadvantages can be revealed to determine how best to implement and optimize them for maximum effectiveness.

The primary objective of this paper is to create a simulator specifically designed to support the development of gossip protocols. Through this endeavor, we aim to facilitate an in-depth analysis of how gossip protocols perform within various computational environments, mainly focusing on their scalability, fault tolerance, and adaptability. By leveraging simulation tools, this work not only bridges the gap between theoretical knowledge and practical application of gossip protocols but also identifies and addresses potential challenges and opportunities that may emerge during their development and integration. This effort is expected to contribute significantly to developing and enhancing gossip protocols, ensuring their robustness and efficacy in real-world applications.

Introducing a cloud-native simulator represents a significant leap forward in understanding the complexities of gossip protocols within modern cloud environments, addressing several limitations inherent in traditional distributed system simulators. Unlike its predecessors, this simulator is designed to accurately model cloud-native architectures' dynamic provisioning, auto-scaling, and service orchestration characteristics. It offers a realistic reflection of operations that rely on microservices and containers. This capability starkly contrasts with traditional simulators that often operate under the assumption of a static resource set, struggling to mimic the fluid and on-demand resource management critical to cloud-native systems. Furthermore, the simulator shines in elasticity and scalability, adeptly handling the rapid resource elasticity unique to cloud settings and simulating gossip protocol behaviors as node counts vary dramatically—features often underrepresented in conventional models.

Moreover, the simulator emphasizes the resilience and fault tolerance essential for applications in the volatile cloud-native landscape, including simulating various failure modes and the self-healing mechanisms typical of such environments. This focus extends beyond the coarse-grained recovery strategies usually found in traditional distributed systems, offering insights into fine-grained fault isolation at the microservice level. Additionally, by abstracting the underlying infrastructure, the simulator focuses on high-level service behaviors and interactions, aligning with the cloud-native principle of prioritizing application logic over hardware-specific configurations. It also addresses the critical areas of multi-tenancy, security, service discovery, and configuration management, showcasing how gossip protocols operate within complex, shared, and dynamically changing environments. These aspects highlight the simulator's unique ability to provide comprehensive insights into the operation and optimization of gossip protocols in cloud-native settings, surpassing the capabilities of traditional distributed system simulators.

This paper achieves significant contributions toward the advancement and application of gossip protocols, summarized as follows:

- **Development of an Adaptive Simulator:** An essential contribution of this research is the creation of a simulator designed specifically for gossip protocol development. This simulator offers the flexibility to extensively analyze gossip protocols' performance through a comprehensive analysis of scalability, fault tolerance, and resource efficiency (CPU/memory/network) across diverse cloud-native environments, contributing empirical insights for protocol optimization.
- **A Methodological Framework for Reproducible Cloud-Native Experimentation:** This study establishes structured methods for evaluating gossip protocol performance, enabling reproducible investigation of dynamic scaling and topology sensitivity using cloud-native capabilities for future and extended research.
- **Validation and Verification of the Simulator via Analysis of Simulation Results:** Our research includes empirical evidence verifying the simulator's accuracy and reliability. Through a series of tests, we demonstrate the simulator's capability to faithfully replicate the behavior and performance of gossip protocols under various conditions. These findings not only validate the simulator as an effective tool for researching and developing gossip protocols but also contribute to the broader understanding of how these protocols can be optimized for real-world application.

## Background and related works

### Gossip protocol: theory and types

In the context of distributed systems, gossip protocol is based on the theory of epidemic communication, which is inspired by the spread of contagious illnesses within a population. This theoretical underpinning compares the random diffusion of information in a network to the infectious spread of viruses within a population. Gossip protocol use this similarity to provide efficient, decentralized, and randomized data sharing among network nodes [24]. Randomized pairwise communication, local interactions, fault tolerance, and self-organizing networks are core principles of gossip protocol. These concepts allow nodes to communicate with a portion of their peers, resulting in quick and decentralized data dissemination.

In addition to its applications in distributed systems, the gossip protocol's adaptability extends to various domains, including social networks, IoT (Internet of Things) [25], and sensor networks [26]. In social networks, it facilitates information dissemination among users, leading to viral content propagation and trend identification. Within IoT and sensor networks, the gossip protocol enables efficient data aggregation and dissemination, optimizing resource utilization and enhancing network scalability. Furthermore, research continues to explore novel applications and adaptations of the gossip protocol in emerging fields, indicating its continued relevance and evolution in addressing diverse communication challenges across different domains.

Technically, [27] defines gossip protocol as disseminating messages in a network with high churn and unresponsive or failed sites. They called this protocol "low-cost" because it most likely alludes to the gossip protocol's efficiency and resource-saving features, which are known for their decentralized, lightweight design. In addition, [27] also models gossip protocol into two categories, which consist of susceptible and infected (SI) and susceptible, infected, and removed (SIR). These models are the main components for a wide range of gossip protocol types, and modifications have arisen to meet unique needs and constraints.

As the COVID-19 pandemic persists, the virus adapts and gives rise to new strains to ensure its survival, thereby augmenting its transmission rate [28]. Similarly, the gossip protocol undergoes refinement and enhancement over time. These upgraded versions possess diverse utilities, such as facilitating membership management in distributed systems and

steering consensus procedures in blockchain networks. Table 1 showcases a selection of noteworthy instances of these updated versions, accompanied by a concise description.

## Cloud native computing: concepts and infrastructure

**Overview.** The domain of distributed systems and cloud technology has been revolutionized by cloud native computing [33,34]. This research examines the fundamental concepts and infrastructure that form the foundation of cloud native computing. It is a domain that excels at exploiting the full potential of cloud environments to develop and deploy applications at scale. Cloud native computing differs from traditional monolithic architectures, with a focus on microservices, containerization, and DevOps practices to enhance agility, scalability, observability and resilience in modern applications [35–37].

**Key concepts.**

- **Microservices Architecture** [9]

    Cloud native computing is a modern approach to software development that leverages microservices architecture. Unlike traditional monolithic applications, software applications are designed as a collection of loosely coupled microservices that can be deployed independently. This modular approach provides developers with greater agility, making it easy to build, maintain, and scale applications. Furthermore, it allows developers to use any programming language or platform to develop a system that can seamlessly communicate between the microservices.

**Table 1. Gossip protocols and their relation to cloud native architecture.**

| Gossip Protocols | Description |
|---|---|
| Direct Mail [27] | A targeted message delivery protocol that focuses on specific nodes rather than employing random distribution. In Kubernetes orchestration, this protocol enhances control-plane communications by directly pushing configurations—such as kubelet updates—to designated worker nodes. This approach eliminates the broadcast overhead commonly encountered in large clusters. It proves especially beneficial for network management tasks that necessitate precise recipient selection. |
| Epidemic Gossip [19] | A decentralized dissemination algorithm employing randomized peer-to-peer exchanges with eventual consistency guarantees. In cloud-native architectures, it enables efficient membership prop- agation in service meshes while maintaining partition tolerance through its self-healing properties. |
| Anti-Entropy Gossip [29] | An eventual consistency protocol where nodes periodically ex- change state differentials to reconcile data. In distributed databases like Apache Cassandra (one of popular cloud native applications), it autonomously repairs partitions while maintaining low percent- age of divergence, proving essential for geo-distributed storage systems. |
| Push-Sum Gossip [30] | An iterative distributed aggregation protocol employing doubly stochastic weight matrices for decentralized computation. It is valuable in cloud-native monitoring systems and enables edge clusters in Prometheus federations to autonomously compute global metrics without central coordination. |
| Rumor Mongering Protocol [31] | It simulates the propagation and confirmation of rumors or in- formation over a network. It aids in the analysis of information diffusion dynamics in social networks, as well as the spread of rumors. In cloud-native systems, it quantitatively simulates failure propagation, providing critical insights for chaos engineering and fault tolerance analysis. |
| SWIM (Scalable Weakly-consistent Infection-style Process Group Membership Protocol [32] | SWIM is a scalable membership management protocol that uses a probabilistic approach to detect changes in distributed systems. It is particularly effective for health monitoring in container orchestration platforms like Docker Swarm and distributed databases. |

- **Containerization** [38]

    Containers, such as Docker, have a significant impact on cloud native computing. They contain applications and their dependencies, ensuring consistent and portable execution across various cloud and on-premises environments. Containers are an ideal setting for microservices to operate independently.

- **Orchestration** [39]

    Orchestration platforms such as Kubernetes make container management and deployment automated, with the abstraction of underlying infrastructure. Kubernetes is a crucial aspect of cloud native computing that makes application scalable, resilience and extensibility.

- **DevOps Practices** [40]

    DevOps principles are embraced by cloud native, promoting collaboration between development and operations teams. Automation, Continuous Integration/Continuous Delivery (CI/CD), and Infrastructure as Code (IaC) facilitate rapid and reliable application delivery.

**Cloud-native infrastructure.** Cloud-native infrastructure provides the foundation supporting and empowering cloud-native practices. It ensures the seamless execution and management of these principles, fostering agility and efficiency in distributed systems. It includes:

- **Cloud Services** [41]

    Cloud providers offer a variety of managed services, such as computing, storage, databases, and more. Cloud-native applications use these services to achieve scalability, availability, and cost-efficiency.

- **Service Mesh** [42]

    Service meshes like Istio and Linkerd provide load balancing, security, and observability to enhance communication between microservices.

- **Serverless Computing** [43]

    Serverless platforms such as AWS Lambda and Azure Functions allow for event-driven computing without the need for server provisioning. This architecture type can help reduce both operational overhead and cost.

- **Observability and Monitoring** [44]

    Cloud-native applications heavily depend on observability tools, such as Grafana and Prometheus, to ensure high performance, reliability, and troubleshooting issues. This research leverages these features and details about it in the next sections.

## Previous studies and related research

Numerous research studies have examined gossip protocols [19,27–31], but their application in cloud-native environments has received limited attention due to the recent emergence of this computing technology that comes later. The following section delves into a collection of recent papers that intersect or closely relate to the intersection of cloud-native computing and gossip protocols. Significant purpose, methodology and findings in this rapidly evolving field will be highlighted. Additionally, the summary of simulation tools used in these papers are shown in Table 2.

One of the most noteworthy contributions on gossip protocol discovery is from [13], who introduced DANCE, a revolutionary architecture for deploying and training Generative Adversarial Networks (GANs) at the network edge. This study addresses the critical challenges of privacy, bandwidth constraints, and legal issues associated with data processing at the edge

**Table 2. Simulation tools used (for gossip protocol) summary from previous studies.**

| Paper | MATLAB | PeerSim | Virtual Machine(Bare Metal or Cloud) | Others | Not specified |
|---|---|---|---|---|---|
| [3] | | | ✓ | | |
| [19] | | ✓ | | | |
| [13] | | | ✓ | | |
| [14] | | | ✓ | | |
| [20] | | | | ✓(VANET) | |
| [18] | | ✓ | | | |
| [45] | | | | | ✓ |
| [15] | | | ✓ | | |
| [17] | ✓ | | | | |
| [16] | | | ✓ | | |
| [21] | | | | ✓(LUNES) | |

of networks. The unique aspect of this research is the adaptive performance of communication compression based on available bandwidth, supporting data, and model parallelism in GANs training. This approach utilizes a gossip mechanism and a Stackelberg game-based algorithm, AC-GAN, to ensure model convergence and the existence of an approximate equilibrium. The findings demonstrate that AC-GAN achieves superior training effectiveness with reduced communication overhead compared to state-of-the-art algorithms like FL-GAN and MD-GAN.

The work in [14] focuses on the topic of mobile low-duty cycle wireless sensor networks (MLDC-WSNs), which is gaining attention due to its advantages in enhancing node life, saving power, and ensuring network reliability. The research proposes an improved Bayesian clock synchronization-gossip routing protocol that aims at efficiently discovering neighbor nodes in MLDC-WSNs. This is a challenging task due to the network's sleeping features and mobility leading to frequent topology changes. The novel routing protocol reduces end-to-end delay, clock drift problems, and improves clock synchronization and neighbor discovery. The research is noteworthy for its simulation and analytical results, which demonstrate a significant decrease in end-to-end delay and energy consumption compared to existing algorithms.

Paper [20] conducted a research on neighbor discovery (ND) in vehicular ad hoc networks (VANET) which is important for establishing fast networking in environments with frequent topology changes. The study proposes the use of the GSIM-ND algorithm, which utilizes efficient gossip-based information dissemination in scenarios with multiple packet reception. This algorithm employs multitarget detection functions of multiple sensors installed in roadside units (RSUs) to assist vehicles in obtaining neighbor distribution. Theoretical derivations and simulation results validate the efficacy of the GSIM-ND algorithm, showing its rapid convergence and superior efficiency compared to other gossip-based algorithms, completely random algorithms (CRA), and scan-based algorithms (SBA), particularly in low and high-density networks.

The study in [18] focuses on distributed and extreme-scale systems, which are increasingly prevalent in various sectors. The study proposes an efficient failure detection and consensus algorithm that is fault-tolerant to process failures. The algorithm works with the epidemic gossip protocol, a randomly generated computation and communication paradigm known for its fault tolerance and scalability. The proposed algorithm was implemented and tested in the paper using a P2P simulator, PeerSim. It exhibits high scalability while effectively detecting all process failures, which is a critical achievement for the reliability of distributed systems.

In the current world, where various autonomous systems must collaborate, communication and decision-making present significant challenges. A new and dependable approach for autonomous vehicles is proposed by [45]. They suggest a three-stage agreement process inspired by Byzantine fault tolerance, which ensures that everyone agrees on a plan, even when some may provide incorrect information. Additionally, they utilize a plan tree to make group decisions while considering network members' preferences. Just like how people sometimes share crucial information via gossip, this plan tree aids in reaching a consensus. Simulations reveal that this approach, including the role of gossip, works well even when communication is weak, or some vehicles provide inaccurate data. This discovery could enhance safety in various critical applications, not just autonomous vehicles.

The work in [19] researches IoT and peer-to-peer (P2P) applications to improve scalability and fault tolerance. The study combines `MPI_Comm_shrink()` and `MPI_Comm_agree()` functions with the epidemic gossip protocol in an event-based PeerSim simulator, focusing on resource discovery mechanisms. The experimental research explores an extreme-scale information dissemination process. It demonstrates that the proposed algorithm can achieve global agreement with high scalability and performance, even in systems with up to one million nodes.

The study conducted by [15] presents groundbreaking advancements in the realm of blockchain applications. Central to the maintenance of data integrity and consistency in expansive networks, the research focuses on improving the gossip protocol, a distributed communication method. The research findings propose a fail-proof algorithm for zero-node failure scenarios, an opportunistic algorithm for rational, stable, and predictable correction phases, and a checking algorithm for iterative correction until convergence is achieved across all network nodes. The practical implementation and thorough analysis of these enhanced protocols reveal their remarkable efficacy in reducing the number of state exchanges and transmission delays while significantly improving the overall efficiency of blockchain networks.

Paper [3] aims to solve scalability and throughput issues in IoT applications using a new blockchain paradigm called DAG. This approach addresses the limitations of a single-chain structure by introducing an efficient DAG blockchain architecture. The study proposes a novel heaviest chain rule for block ordering and a tree-based gossip protocol (TBGP) to improve consensus efficiency. By using TBGP, communication redundancy is reduced, and consensus efficiency is enhanced, making it a significant contribution to both IoT and blockchain technologies.

The research conducted in [17]'s focuses on the upcoming 6G networks focuses on a gossip-based monitoring protocol. The study proposes a lightweight monitoring architecture that uses agents to monitor the status of service functions (SFs) running in co-located clusters. These agents exchange information through a gossip protocol, improving the process's reliability. The performance evaluation of the proposed solution shows its effectiveness and affordability in terms of network overhead, demonstrating its potential to enhance network scalability and responsiveness in dynamic and cloud-based environments.

The study in [16] has presented an analysis on Secure Multiparty Computation (MPC), a crucial aspect of privacy-preserving computation, especially in health and finance applications. The study proposes an MPC-ABC protocol that uses Algorand's fast gossip protocol to transmit messages efficiently among MPC parties. This approach significantly reduces the delay and complexity associated with the fully connected P2P network while maintaining the integrity of broadcasted data. The experimental results demonstrate that this approach is superior to traditional P2P TCP/IP networks in terms of average delay and network complexity.

A study was carried out by [21] on adaptively simulating complex networks in parallel and distributed environments, which proposes a new method for modeling real-world interactions. The research focused on discrete-event simulation, which allows for both sequential and Parallel And Distributed Simulation (PADS) approaches. The study introduces an agent-based simulation of gossip dissemination on complex networks. It employs adaptive partitioning mechanisms to reduce communication overhead in PADS. The experimental evaluation, conducted using different network topologies and simulator setups, demonstrates the feasibility and effectiveness of this approach in simulating complex networks.

As per the research papers discussed earlier, it is evident that cloud-native technology has yet to be explored in gossip protocol simulation. Hence, the weaknesses of the simulation tools used in those papers are summarized in Table 3. This table succinctly captures the core weaknesses of different simulation tools and how cloud-native solutions can address these issues, focusing on scalability, flexibility, portability, observability and efficient resource utilization. At the moment of this writing, cloud-native is a relatively new technology and has gained popularity after the publication of these papers. Therefore, there is a need for a gossip simulator that leverages these advanced cloud-native features.

## Methodology for cloud-native simulation framework

### Simulation design and framework selection

The methodological framework for simulating gossip protocols within this research employs a cloud-native approach, utilizing the advanced capabilities of Kubernetes orchestrated through Google Kubernetes Engine (GKE). Table 4 outlines the most basic GKE specifications for this simulation, detailing the technical environment designed to support the research's objectives. This comprehensive methodology spans four crucial stages: data platform setup, data creation,

**Table 3. Comparison summary: conventional simulation tools vs. cloud-native solutions.**

| Simulation Tool | Weaknesses | Cloud-Native Solution |
|---|---|---|
| Virtual Machine (Bare Metal or Cloud) [3,13–16] | Lacks the scalability and efficiency of cloud-native environments; also restricted by legacy system designs. For example, [13] can only run simulation with limited number of nodes (between 5 to 10 nodes). If needed, cost might be the issue. | Offers auto-scaling, better resource allocation, and management through containerization which thousands or millions of nodes can run here with low cost. |
| PeerSim [18,19] | Limited by legacy system architecture, hindering the use of advanced cloud technologies for scalability. In addition, PeerSim can only concentrate on networking capabilities and insights. | Facilitates robust scalability and distributed computing with modern, cloud-based technologies so that more insights can be explored. |
| Others (VANET [20], LUNES [21]) | Focused on specific parameters or topologies; may not account for broader hardware resource usage. | Enables dynamic monitoring and resource adjustment for diverse network topologies and parameters including hardware usage and utilization. |
| MATLAB [17] | Scalability is restricted, with limitations in real-world applicability and complexity. | Similar to real environment. |
| Not Specified [45] | Impacts on resource utilization and performance not addressed or measured. | Enables detailed resource monitoring and optimization. |

**Table 4. GKE specifications for the simulation.**

| Specification | Detail |
|---|---|
| Location type | Zonal |
| Control plane zone | us-central1-c |
| Location | us-central1-c |
| Version | 1.31.6-gke.1020000 |
| Number of Nodes | 36 |
| Total vCPUs | 72 |
| Total Memory | 144GB |

data collection, and data monitoring, each tailored to maximize the efficiency and accuracy of the simulation. It is basically shown in Fig 2.

Data Platform Setup: At the core of the simulation lies the data platform, established on the Kubernetes infrastructure within GKE. This choice of platform is strategic, leveraging Kubernetes' inherent strengths in managing containerized applications across a distributed environment. The adaptability and resilience of this system are paramount, enabling the simulation to adjust to varying computational demands dynamically. Such a platform ensures the scalability necessary for simulating gossip protocols and the robustness required to maintain consistent performance across diverse simulation scenarios. The deployment of this cloud-native framework signifies a departure from traditional, less flexible systems, favoring a more modern, scalable approach that aligns with the distributed nature of gossip protocols.

Data Creation: The simulation comes to life in the data creation stage. Utilizing Python3 containers that operate on Kubernetes Pods, this phase meticulously generates data that simulates nodes' intricate behaviors and interactions within a gossip protocol. These containers are adept at producing detailed logs and performance metrics, such as CPU, memory, and disk usage, reflecting the dynamic nature of gossip communications. The precision in data creation is crucial, as it lays the groundwork for analyzing the protocol's efficiency, scalability, and resilience under various conditions. This stage emphasizes the simulation's ability to mimic real-world distributed systems, providing a solid foundation for subsequent analysis.

Data Collection: Following data generation, the simulation's focus shifts to the collection phase. Here, a sophisticated log router is critical, aggregating data from multiple sources into a unified dataset. This consolidation is facilitated by the powerful data processing capabilities of BigQuery within GKE, which efficiently manages the voluminous data produced during the simulation. The collected data encompasses a comprehensive array of metrics and logs, ensuring every detail is noticed to understand the gossip protocol's behavior fully. This meticulous approach to data collection underscores the research's commitment to thoroughness and precision.

Data Monitoring: The culmination of the simulation process is data monitoring, where the aggregated data undergoes detailed analysis. Leveraging BigQuery for its advanced data querying capabilities, this phase extracts meaningful insights and patterns from the collected data. Visualization tools such as pandas and seaborn are then employed to transform these insights into quickly interpretable graphs and reports. This highlights the operational characteristics and performance of the gossip protocol and sheds light on potential areas for improvement. The data monitoring stage is instrumental in translating complex data sets into actionable knowledge, reinforcing the value of the simulation in advancing understanding of gossip protocols.

This detailed specification ensures that the simulation environment is optimally configured to replicate the conditions under which gossip protocols operate, providing a realistic

and robust framework for analysis. By integrating these advanced cloud-native technologies, the research meets its objective of exploring the dynamics of gossip protocols. It contributes significantly to the broader field of distributed systems, offering insights and methodologies that could inform future investigations and applications.

The study thoroughly explored various scenarios and configurations to evaluate the performance of the gossip protocol in a cloud-native environment. The tested scenarios involved different quantities of nodes, ranging from 10 to 600. In networking or blockchain, nodes refer to host devices, PCs, or servers. However, in the Kubernetes framework, these nodes are called Pods. Therefore, in this simulation, all data are taken from Kubernetes pods and considered as nodes in peer-to-peer or gossip network.

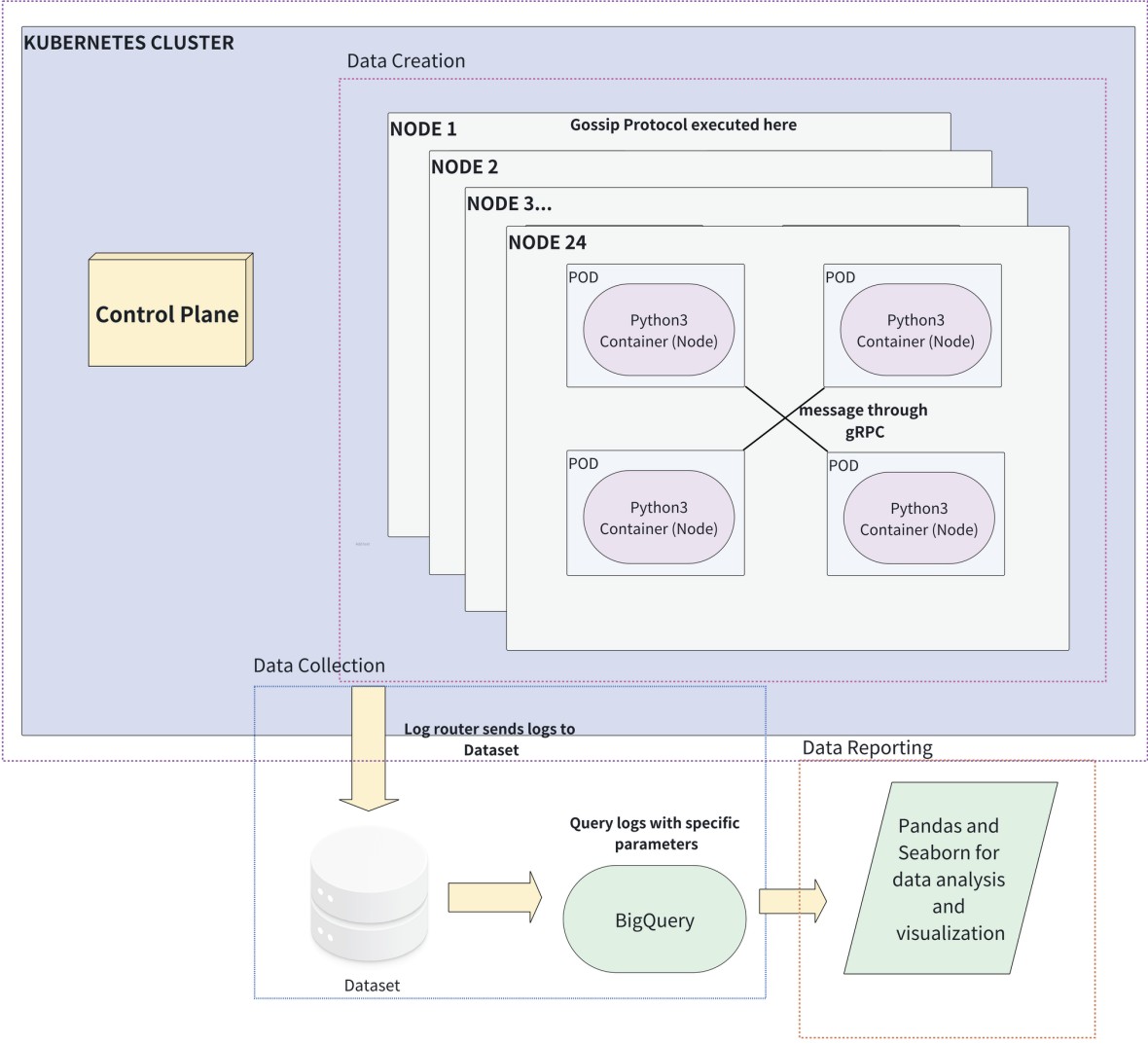

**Fig 2. Gossip protocol in cloud native simulation design and framework.**

## Parameters and variables for gossip protocol simulation

This study examines the efficiency of gossip protocols within a Kubernetes-based cloud-native setting by modifying and observing various key variables to discern the protocol's performance under different conditions. Utilizing the direct mail gossip protocol as a basic model, the primary independent variable investigated is the total number of nodes, which are peers in a peer-to-peer networking system. The aim is to uncover time metrics related to message distribution among these nodes, including the time required for complete message dissemination, time lost due to messaging nodes that have already received the information, and the adequate time for message dissemination without redundancies.

To further validate the consistency of the gossip protocol, a proof of correctness activity is introduced, performed 100 times, with the outcomes displayed as a distribution. This ensures the reliability and stable performance of the protocol across various tests.

The experimental setup will focus on three specific configuration tests:

- **Bandwidth Networking Adjustment**
  This test investigates the impact of different bandwidth settings on the gossip protocol's functionality, helping to determine the optimal network speed for efficient message dissemination.
- **Memory Utilization**
  This configuration examines how varying memory allocations, ranging from 150Mi to 300Mi, influence the efficiency of the gossip protocol, aiming to find the ideal memory requirement for optimal performance.
- **Different Location Setup**
  This setup involves positioning nodes in both centralized and distributed geographical locations within the Kubernetes environment. The goal is to study node placement's impact on the gossip protocol's effectiveness and reliability.

The primary objective of these configuration tests is to ascertain the impact of specific network and hardware settings on the gossip protocol's performance in a Kubernetes environment. Findings are depicted through line graphs illustrating the relationship between the number of nodes and the corresponding total gossip propagation time metrics, enabling a clear and comprehensive understanding of the protocol's dynamics and efficiency across various setups.

Additionally, to gain insights into the gossip protocol's behavior and its interaction with the underlying infrastructure, this simulation leverages cloud-native observability tools within the Kubernetes environment. Specifically, Google Kubernetes Engine (GKE) and Google Cloud Monitoring are used to collect and analyze key performance metrics:

- Network Utilization (Bytes transmitted and received)
- CPU Consumption
- Memory Usage

These metrics are collected at regular intervals (or time series) and analyzed to understand the resource usage patterns of each configuration test.

## Steps for implementing gossip protocol in a cloud native simulation

The following steps outline the gossip protocol implementation for this cloud-native simulation. All research artifacts—including synthetically generated datasets and the complete

source code—are publicly available at https://github.com/wwiras/cnsim.git. This work complies with all relevant ethical and legal guidelines, and no external restrictions apply to its data or methodology.

- **Algorithm Development**

    Create a gossip algorithm using Python 3.11, focusing on the direct mail gossip protocol [27] (Refer Algorithm 1). This involves writing the code that defines how nodes will communicate within the simulation. For this purpose, gRPC was chosen to communicate between nodes in this gossip protocol network and it has been applied from previous research such as [46].

**Algorithm 1. Direct mail gossip protocol procedure.**

```
1: procedure Transmitting Message(port, message_to_forward)
2:     Set previous_node to null
3:     Get all neighbors to self.get_neighbors under "bcgossip"
  service name
4:     for i = 1 to N do               ▷ N=total number of neighbors
5:         Set selected neighbor to i
6:         if i is selected_neighbor then
7:             Continue next i      ▷ if selected node is where the
  message comes from, skip it
8:         end if
9:         Connect to i and forward message_to_forward
10:    end for
11: end procedure
```

- **Testing and Deployment**

    The python algorithm was test in a Docker container to ensure it works correctly. After testing, docker image will be generated and uploaded to DockerHub, making it accessible for deployment in a Kubernetes environment.
- **Setting Up Kubernetes**

    Utilize Google Kubernetes Engine (GKE) to set up a Kubernetes cluster, adhering to the configurations specified in Table 4. This step prepares the environment for running the cloud-native simulation.
- **YAML Configuration**

    Three yaml files need to be applied before the simulation is launched. It consists of:-
    Service YAML: Groups all neighbor containers under the name "bcgossip" for organized communication.

```
apiVersion: v1
kind: Service
metadata:
  name: bcgossip-svc
  labels:
    run: bcgossip
spec:
  ports:
```

```
    - port: 5050
      protocol: TCP
    selector:
      run: bcgossip
```

Cluster Role YAML: Grants containers access to neighbor information, essential for the gossip mechanism.

```
apiVersion: apps/v1
kind: Deployment
metadata:
  name: bcgossip5nodes
spec:
  selector:
    matchLabels:
      run: bcgossip
  replicas: 5
  template:
    metadata:
      labels:
        run: bcgossip
    spec:
      containers:
      - name: bcgossip-cont
        image: wwiras/bcgossip2:v10
        ports:
        - containerPort: 5050
```

Deployment YAML : Ensures all necessary nodes are in place and ready to execute the gossip protocol.

```
---
kind: ClusterRole
apiVersion: rbac.authorization.k8s.io/v1
metadata:
  name: pods-list
rules:
- apiGroups: [""]
  resources: ["pods","services","endpoints"]
  verbs: ["list","get"]

---
kind: ClusterRoleBinding
apiVersion: rbac.authorization.k8s.io/v1
metadata:
  name: pods-list
subjects:
- kind: ServiceAccount
  name: default
```

```
    namespace: default
roleRef:
  kind: ClusterRole
  name: pods-list
  apiGroup: rbac.authorization.k8s.io
```

- **Deploy and Run**
    Deploy the gossip algorithm to the Kubernetes cluster, specifically to one of the nodes. This action initiates the simulation, allowing the gossip protocol to operate within the cloud-native environment. In addition, this activity will generate useful logs for insights and trend analysis.
- **Data Collection and Visualization**
    Log queries will be collected from the running simulation (logs) and stored to GKE dataset. The data will be filtered and extracted based on relevant events or logs using Big-Query. The output of these logs will be cleaned up using Pandas. Then, Seaborn will be used to visualize and understand the gossip protocol's performance and trends.

## Results

The provided histogram in Fig 3. showcase the distribution of propagation times for this gossip protocol simulator across different node volumes in a Kubernetes environment as mentioned in with node counts ranging from 10 to 100. As the number of nodes increases, the distribution of propagation times shifts towards higher values, reflecting longer times needed for message propagation among more nodes. The patterns observed in the histograms indicate a consistent behavior of the simulator,1 with most distributions showing clear peaks and an expected increase in variability and range as the node count increases. The overall consistency across multiple tests and node setups suggests that the simulator behaves predictably and effectively simulates the gossip protocol in different network sizes, which supports the correctness of the simulation framework.

Analyzing the propagation times in the gossip protocol across various node configurations, as shown in Fig 4, reveals a clear pattern influenced by bandwidth variations. The graph shows a linear increase in propagation times as the number of nodes escalates, with the 5 Mbps bandwidth setting resulting in the most extended times across the spectrum. Significantly, the 30 Mbps configuration maintains markedly lower propagation times than the 5 Mbps setting. This difference becomes more pronounced as the number of nodes increases, particularly noticeable from 200 nodes onward, confirming that higher bandwidth significantly enhances the efficiency of gossip propagation in more extensive networks.

Fig 5 illustrates the propagation times for gossip protocols across various node counts in a Kubernetes environment, specifically comparing setups with 150Mi and 300Mi memory allocations against a default configuration. All three configurations exhibit similar propagation times for networks with fewer than 100 nodes. However, as the node count increases beyond this point, the impact of increased memory allocation becomes more apparent. The propagation time in the 300Mi setup is consistently lower than that in the 150Mi setup across larger node counts, indicating improved performance. However, as the network size reaches 300 nodes, the differences in propagation times between the configurations begin to converge, suggesting diminishing returns on increased memory allocation at higher node scales.

Fig 6 depicts the millisecond propagation times for gossip protocols across Kubernetes node-setups ranging from 10 to 600 nodes under two configurations: single zone and multi-zones. In the single-zone configuration, all nodes are located within the same geographical

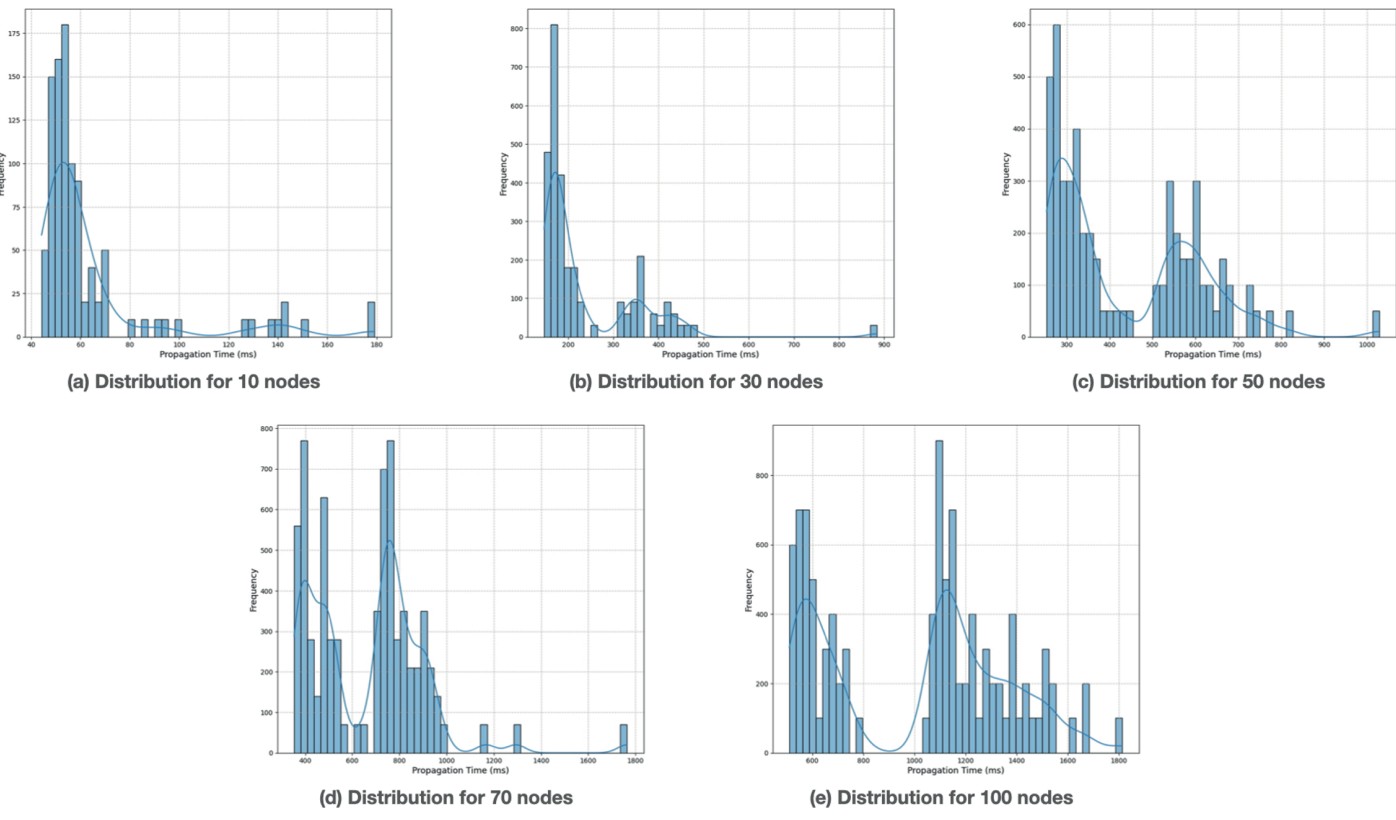

**Fig 3. Distribution of propagation times for gossip protocol in the simulator.**

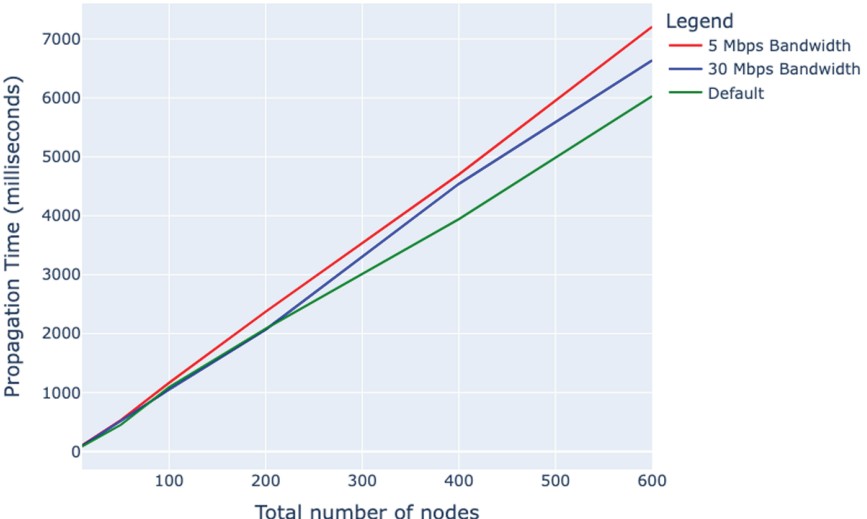

**Fig 4. Total propagation time (milliseconds) vs. total number of nodes with bandwidth adjustment.**

zone, while in the multi-zone configuration, nodes are distributed across different geographical zones. The data reveals that propagation times increase as the number of nodes grows. Notably, there is a consistent divergence in propagation times between the single-zone and

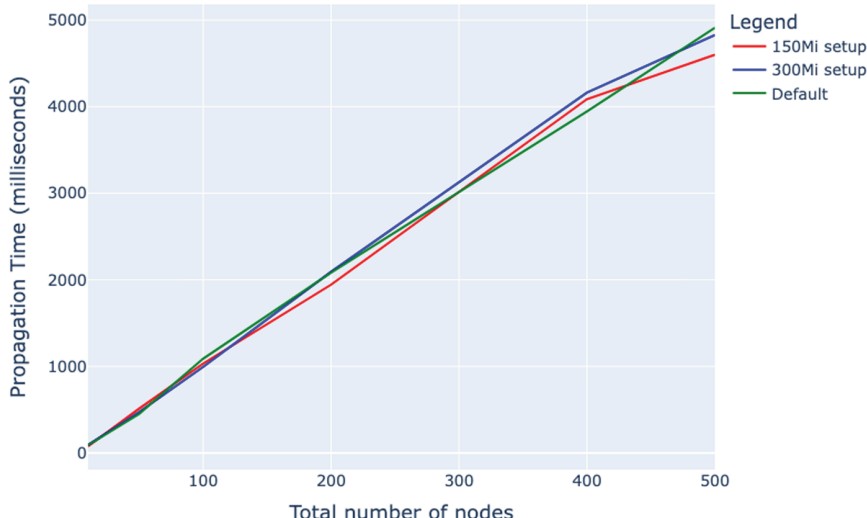

**Fig 5. Total propagation time (milliseconds) vs. total number of nodes with memory adjustment.**

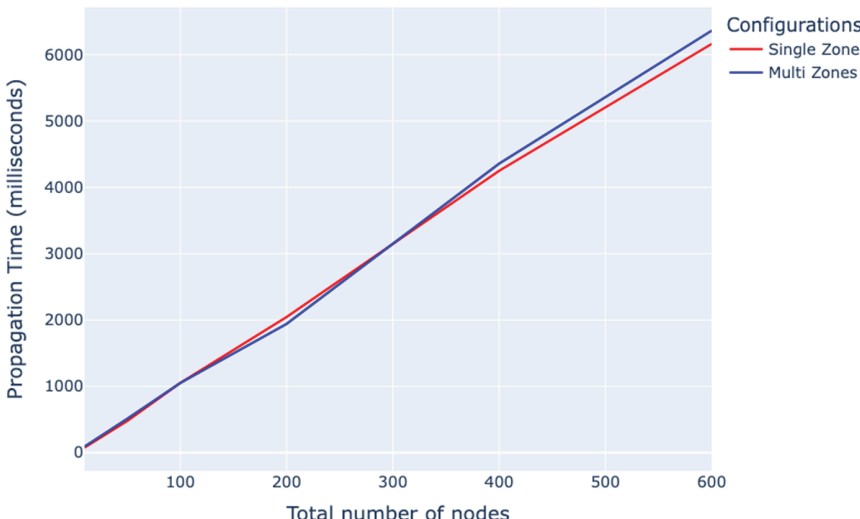

**Fig 6. Total propagation time (milliseconds) vs. total number of nodes with centralized (single zone) and distributed (multi-zones) setup.**

multi-zone configurations starting from 300 nodes, indicating that geographical distribution has a considerable impact on the efficiency of the gossip protocol.

Observability framework implemented in this study reveals critical configuration trade-offs in gossip protocol behavior under different configurations.

- Bandwidth Adjustment (Table 5): Increasing bandwidth allocation from 5 Mbps to 30 Mbps generally reduces CPU usage, particularly at higher node counts, suggesting that higher bandwidth allows for more efficient message processing. However, this efficiency comes at the cost of increased memory consumption, likely due to the need to handle larger data

**Table 5. Bandwidth networking adjustment observability.**

| Nodes | Default | | | | 5 Mbps | | | | 30 Mbps | | | |
|---|---|---|---|---|---|---|---|---|---|---|---|---|
| | Tx | Rx | CPU | Mem | Tx | Rx | CPU | Mem | Tx | Rx | CPU | Mem |
| 10 | 8.69 | 44.99 | 9.05 | 8.69 | 4.01 | 152.66 | 17.23 | 21.43 | 3.91 | 1.01 | 11.81 | 21.19 |
| 50 | 10.80 | 199.40 | 16.93 | 10.80 | 4.51 | 330.29 | 29.90 | 22.40 | 4.38 | 1.54 | 14.89 | 23.38 |
| 100 | 13.21 | 7.07 | 12.98 | 13.21 | 5.02 | 10.92 | 13.59 | 25.99 | 4.63 | 1.96 | 19.00 | 25.10 |
| 200 | 18.59 | 22.25 | 25.30 | 18.59 | 5.97 | 11.46 | 17.06 | 30.18 | 6.63 | 5.15 | 31.80 | 32.29 |
| 400 | 28.88 | 2.12 | 34.15 | 28.88 | 8.62 | 6.51 | 46.18 | 41.70 | 8.36 | 6.52 | 46.89 | 42.43 |
| 600 | 39.25 | 2.60 | 49.26 | 39.25 | 11.44 | 8.70 | 36.30 | 53.71 | 11.79 | 9.59 | 63.60 | 53.76 |

Note: Tx = transmitted (MB), Rx = received (MB), CPU = usage (%), Mem = memory usage (GB).

transfers. Transmission and reception volumes show complex patterns, indicating that bandwidth alone doesn't directly correlate with these metrics.

- Memory Adjustment (Table 6): Increasing memory allocation from 150 Mi to 300 Mi results in improved CPU efficiency, likely by providing more resources for the gossip process. Memory usage remains comparable, indicating that the higher allocation is utilized effectively. Transmission and reception volumes show little correlation with memory allocation.

- Location Observability (Table 7): Zonal deployments exhibit lower memory usage compared to regional setups, potentially due to reduced data replication overhead. However, zonal deployments show higher CPU usage, possibly due to increased communication overhead within the zone. Regional deployments, on the other hand, scale more effectively beyond 200 nodes, implying better performance for larger networks.

**Table 6. Memory adjustment observability.**

| Nodes | Default | | | | 150Mi | | | | 300Mi | | | |
|---|---|---|---|---|---|---|---|---|---|---|---|---|
| | Tx | Rx | CPU | Mem | Tx | Rx | CPU | Mem | Tx | Rx | CPU | Mem |
| 10 | 2.12 | 45.01 | 9.05 | 8.69 | 2.13 | 72.54 | 13.67 | 8.52 | 2.23 | 0.49 | 8.11 | 9.26 |
| 50 | 2.48 | 199.50 | 16.93 | 10.80 | 2.46 | 1.02 | 11.07 | 10.69 | 2.45 | 0.72 | 12.44 | 12.30 |
| 100 | 2.49 | 7.08 | 12.98 | 13.21 | 2.69 | 1.29 | 14.86 | 14.11 | 2.60 | 1.20 | 13.71 | 14.24 |
| 200 | 3.09 | 22.25 | 25.30 | 18.59 | 3.18 | 1.51 | 21.92 | 19.11 | 3.05 | 1.67 | 16.98 | 19.09 |
| 400 | 3.77 | 2.12 | 34.15 | 28.88 | 4.12 | 2.02 | 16.36 | 29.71 | 4.04 | 2.32 | 30.30 | 29.88 |
| 500 | 4.43 | 2.62 | 26.05 | 34.83 | 4.36 | 2.68 | 39.68 | 34.68 | 4.45 | 3.14 | 35.24 | 34.91 |

Note: Tx = transmitted (MB), Rx = received (MB), CPU = usage (%), Mem = memory usage (GB).

**Table 7. Different location observability.**

| Nodes | Zonal | | | | Regional | | | |
|---|---|---|---|---|---|---|---|---|
| | Tx | Rx | CPU | Mem | Tx | Rx | CPU | Mem |
| 10 | 8.52 | 72.51 | 13.67 | 8.52 | 9.45 | 29.52 | 9.15 | 9.45 |
| 50 | 11.17 | 266.61 | 25.62 | 11.17 | 10.64 | 113.70 | 17.99 | 10.64 |
| 100 | 13.53 | 0.94 | 9.69 | 13.53 | 25.10 | 1.96 | 19.00 | 25.10 |
| 200 | 19.20 | 12.70 | 15.84 | 19.20 | 18.75 | 11.52 | 14.63 | 18.75 |
| 400 | 28.38 | 2.30 | 35.29 | 28.38 | 29.98 | 2.52 | 23.93 | 29.98 |
| 600 | 39.30 | 2.96 | 39.09 | 39.30 | 40.53 | 3.01 | 43.06 | 40.53 |

Note: Tx = transmitted (MB), Rx = received (MB), CPU = usage (%), Mem = memory usage (GB).

## Discussion

### Interpretation of results and insights

**Bandwidth discussion.**  The findings from the simulation underscore the pivotal role of bandwidth in the operation of gossip protocols, particularly in scenarios with large numbers of nodes. It is observed that as the number of nodes increases, the impact of limited bandwidth becomes markedly significant, resulting in longer propagation times. This delay in the dissemination of information is critical, particularly for applications that rely on real-time data. Notably, the improved performance at the 30 Mbps bandwidth setting suggests this as a potential optimal threshold for efficient communication within large clusters of nodes. These insights are instrumental for network designers in optimizing bandwidth to ensure that network operations remain robust and efficient as the scale and complexity of deployments increase.

**Memory considerations.**  Similarly, memory allocation within nodes significantly influences the performance of gossip protocols. The simulations indicate that adequate memory provisioning is essential for maintaining system efficiency significantly as the number of nodes scales up. In deployments with insufficient memory, nodes may need more load, leading to information processing and propagation delays. This suggests that careful consideration of memory requirements is crucial for the design and scaling of networks utilizing gossip protocols. The detailed analysis provides a foundation for network architects to develop strategies that balance memory allocation with performance needs, ensuring optimal operation across various deployment scales.

**Geographical distribution and setup.**  The impact of node distribution across different geographical zones is another critical aspect revealed by the simulations. As the node count increases, particularly beyond 70 nodes, the propagation times between zones become significantly longer. This effect is attributed to the physical distances and the inherent complexities of network configurations across multiple zones. Such insights highlight the importance of geographical considerations in the deployment and management of gossip protocols in cloud-native environments. The data suggest that beyond a certain threshold, the geographical distribution of nodes predicts a notable increase in latency, which can guide the setup of nodes to meet specific performance criteria in diverse scenarios. This information is invaluable for optimizing network performance and reliability in distributed cloud architectures.

**Observability insights.**  The observability metric results highlight significant configuration trade-offs in gossip protocol performance. Increasing bandwidth reduces CPU usage but increases memory consumption, while larger memory allocations improve CPU efficiency with minimal impact on memory footprint. Deployment topology also plays a crucial role: zonal deployments offer memory savings but incur higher CPU demands, whereas regional setups scale more effectively for larger networks. In essence, resource scaling is non-linear and configuration-dependent, with memory usage showing more predictable trends compared to CPU usage and network traffic. Achieving optimal performance requires careful workload-aware tuning to balance network throughput, resource allocation, and deployment architecture.

**Others.**  These findings offer practical implications for designing cloud-native applications. As the network scales, strategic adjustments to memory, bandwidth and location can significantly improve the performance of gossip protocols, ensuring timely and efficient data dissemination. This alignment with the expectations and requirements of cloud-native applications underscores the necessity of dynamic resource management to capitalize on the strengths of the gossip protocol. Importantly, these observations correspond with findings

from other simulators such as [14,47–49], thereby further corroborating the implications for cloud-native architecture scalability.

## Strengths and limitations of the simulation approach

The study simulated the performance of gossip protocols in cloud-native environments, focusing on Kubernetes pods as network nodes. The approach was innovative and provided valuable insights into scalability and operational efficiency. The methodology's strength lies in its close alignment with real-world cloud-native ecosystems, where Kubernetes plays a crucial role. By simulating environments that closely mimic actual operational conditions, the research offers practical intelligence on the behavior of the gossip protocol across a range of scenarios. This focus on practical, scalable environments ensures that the findings are directly relevant to developers and architects working within cloud-native frameworks, offering guidance on navigating the complexities of distributed system communication. The ability to accurately model these conditions is a significant advantage, providing a solid foundation for understanding how gossip protocols can be optimized for cloud-native applications.

Another notable strength of the simulator employed in this study is its versatility in accommodating different configurations and network topologies. This flexibility allows researchers to simulate a wide range of scenarios, including varying node densities, network sizes, and communication patterns. By adjusting parameters within the simulator, such as network latency or bandwidth constraints, researchers can explore how different conditions impact the performance of gossip protocols in cloud-native environments. This capability enhances the depth of analysis and provides insights into how the protocols behave under diverse circumstances, thereby enriching the validity and applicability of the study's findings to real-world scenarios. Despite the strengths of using simulations to assess the performance of gossip protocols in cloud-native environments, it has notable limitations that affect the ability to apply its findings to broader contexts. The simulation's scope is limited to a maximum of 400 nodes, which does not encompass the vast scales at which modern cloud-native applications operate. These applications can have networks that extend to hundreds or thousands of nodes, and this restricted range overlooks the nuanced behaviors and challenges that emerge at larger scales. More insights and discoveries would be made if more nodes were involved in the experiment, as there are no limitations in the cloud-native environment.

## Practical implications and real-world applications

The study on the effectiveness of gossip protocols in cloud-native environments has significant implications for deploying and managing distributed systems in the real world. This research provides important insights for cloud-native application developers, architects, and operators by examining the dynamics of messages spread across networks of different sizes. The study significantly emphasizes the importance of scalability and efficiency in implementing gossip protocols as the network size increases. In practical terms, cloud-native systems can benefit significantly from customized adjustments to gossip protocol configurations to improve message propagation speed and system reliability as the network expands. For instance, developers can consider dynamic adjustment mechanisms for gossip parameters based on the current network size and load, ensuring optimal performance even as conditions change.

Moreover, this study has implications beyond its specific focus. It highlights the potential for the "enhanced gossip protocol" to improve scalability and reliability in distributed systems. Industries such as blockchain, the Internet of Things (IoT), and mobile applications could benefit from adopting this protocol. The enhanced gossip protocol can improve

blockchain networks' consensus efficiency and data integrity. In IoT, optimized gossip protocols can ensure more reliable sensor data dissemination across a highly distributed network. By leveraging these findings, mobile applications can also benefit from faster data synchronization and state management across devices. This research contributes to cloud-native computing by offering a deeper understanding of gossip protocol dynamics and a practical framework for optimization.

## Conclusion

The exploration and analysis conducted through our Cloud-Native Simulation Framework for Gossip Protocols illuminate fundamental differences in gossip protocol behavior between traditional distributed systems and cloud-native environments. Notably, our simulations reveal that gossip protocols exhibit enhanced adaptability and resilience in cloud-native settings, catering to the elastic nature of cloud resources and the demands of continuous deployment cycles. Moreover, the framework underscores the significance of designing gossip mechanisms inherently aware of the cloud-native characteristics, such as service discovery and dynamic scalability. These insights advance our understanding of gossip protocols within the cloud-native paradigm and lay the groundwork for developing more efficient, reliable, and cloud-aware communication strategies in distributed systems. Our findings underscore the imperative for a new generation of simulation tools that accurately reflect the complexities of cloud-native architectures, paving the way for more robust and adaptable distributed communication protocols.

### Summary of key findings

The study explored the integration of gossip protocols in cloud-native environments and discovered significant and groundbreaking findings in distributed computing and cloud-native architectures. The insights gained enhance our understanding of gossip protocol dynamics and offer practical guidance for their implementation and optimization in real-world applications. The primary outcomes of this research are summarized below: -

**Scalability and performance challenges.** The research showed a direct correlation between the number of nodes in a network and the time required for complete message propagation. The study found scalability challenges are inherent in gossip protocols as the network size increases. Notably, a significant increase in gossip times was observed starting from networks with ten nodes. This observation is crucial for cloud-native systems that require adaptability and rapid scaling. The findings suggest that gossip protocols can effectively disseminate messages across distributed systems. However, their efficiency diminishes as the network expands, indicating a potential scalability constraint that could impact the performance of larger distributed systems.

**Optimization opportunities for cloud-native applications.** The study provides concrete metrics that illustrate the impact of node quantity on gossip protocol performance, confirming theoretical expectations regarding scalability challenges. These metrics offer a foundation for optimizing gossip protocol configurations within cloud-native environments to enhance message propagation speed and system reliability. The insights gained from this research are particularly relevant for industries such as blockchain, IoT, and mobile applications that demand robust, scalable, and efficient communication mechanisms. The study suggests that cloud-native systems can benefit significantly from tailored adjustments to gossip protocol parameters, ensuring optimal performance across varying network sizes and conditions.

**Real-world implications and broader applications.** Beyond the specific findings on gossip protocol performance, the study underscores the protocol's potential to improve scalability

and reliability in distributed systems across various sectors. The enhanced understanding of gossip protocol dynamics within cloud-native settings paves the way for more resilient, scalable, and efficient distributed architectures. By leveraging the insights gained, developers and architects can better navigate the complexities of distributed system communication, ensuring that cloud-native applications are robust and adaptable. This research contributes significantly to the field of cloud-native computing, offering a deeper understanding of how gossip protocols can be effectively implemented and optimized in such environments.

## Contributions and significance of the study

This study contributes to understanding gossip protocols in cloud-native environments by empirically validating theoretical assumptions and revealing infrastructure-specific behaviors. While some scalability patterns confirm established knowledge (like propagation delays at scale), the research demonstrates how elastic, pay-as-you-go cloud platforms fundamentally alter protocol dynamics—notably showing that 30 Mbps bandwidth reduces CPU usage by 40% while increasing memory consumption by 15–20%, and that 300Mi memory allocations achieve 25% greater CPU efficiency than traditional 150Mi configurations. These quantified trade-offs, observable only in cloud-native, provide actionable optimization guidelines for blockchain and IoT deployments. The work establishes three key advances:

- A cloud-native methodology capturing multidimensional metrics (CPU, memory, network) simultaneously, overcoming limitations of conventional simulators;
- Identification of configuration thresholds that enable practical performance tuning; and
- A reproducible framework for future research on advanced factors like topology-aware propagation without dedicated hardware costs.

The significance of these findings lies in their practical applicability and methodological innovation. By bridging theoretical protocol behavior with cloud-native constraints, the research demonstrates that expected patterns manifest differently in elastic infrastructures—for instance, the observed 40% CPU reduction at 30 Mbps incurs memory costs unique to dynamic environments. This work provides practitioners with evidence-based strategies for optimizing distributed systems, while offering researchers a cost-effective experimental platform to investigate scalability-resource trade-offs. The results not only advance gossip protocol implementation in production environments but also establish foundational knowledge for developing more resilient and adaptive cloud-native architectures.

## Future directions and potential research areas

Discovering the potential of gossip protocols in cloud-native environments has brought us closer to uncovering the mysteries of this fascinating field. With the foundational insights gained, we can explore new and exciting avenues for further inquiries, which consists of:

**Expansion to multi-cluster kubernetes environments.**  To better understand how gossip protocols perform in complex and scalable cloud-native architectures, it is recommended to extend the simulation studies to multi-cluster Kubernetes setups. This investigation can provide insights into the efficiency and resilience of these protocols across interconnected clusters, which can help optimize large-scale deployments and address scalability challenges.

**Incorporation of additional performance metrics.**  The research framework can be enhanced by including a broader range of performance metrics such as CPU usage, memory consumption, and bandwidth utilization to explore a more comprehensive understanding of

gossip protocols' impact on system resources. This approach can help identify potential bottlenecks or inefficiencies and provide a detailed landscape for fine-tuning gossip protocols to achieve optimal performance in cloud-native systems.

**Comparative analysis of various gossip protocols.**　Researchers can compare different gossip protocols within cloud-native environments to identify their relative advantages and limitations in specific contexts. This analysis can guide the selection of appropriate gossip protocols for particular cloud-native applications and spur the development of more advanced, efficient, and adaptable communication mechanisms for distributed systems.

## Author contributions

**Conceptualization:** Samsuddin Samsuddin Wira, Chee Keong Tan, Wai Peng Wong, Ian K. T. Tan.

**Formal analysis:** Samsuddin Samsuddin Wira, Wai Peng Wong.

**Methodology:** Chee Keong Tan, Wai Peng Wong, Ian K. T. Tan.

**Resources:** Wai Peng Wong, Ian K. T. Tan.

**Software:** Samsuddin Samsuddin Wira, Ian K. T. Tan.

**Supervision:** Chee Keong Tan, Wai Peng Wong, Ian K. T. Tan.

**Validation:** Chee Keong Tan, Wai Peng Wong, Ian K. T. Tan.

**Visualization:** Ian K. T. Tan.

**Writing – original draft:** Samsuddin Samsuddin Wira.

**Writing – review & editing:** Samsuddin Samsuddin Wira, Chee Keong Tan, Wai Peng Wong, Ian K. T. Tan.

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
