## [Decision Letter · Decision Letter 0]

19 Feb 2025

PONE-D-24-43101Cloud-Native Simulation Framework for Gossip Protocol: Modeling and Analyzing Network DynamicsPLOS ONE

Dear Dr. Samsuddin,

Thank you for submitting your manuscript to PLOS ONE. After careful consideration, we feel that it has merit but does not fully meet PLOS ONE’s publication criteria as it currently stands. Therefore, we invite you to submit a revised version of the manuscript that addresses the points raised during the review process.

We look forward to receiving your revised manuscript.

Kind regards,

Jacopo Soldani

Academic Editor

PLOS ONE

3. In your Methods section, please include additional information about your dataset and ensure that you have included a statement specifying whether the collection and analysis method complied with the terms and conditions for the source of the data.

“Internal Matching Grant - Public Service Department (JPA) of Malaysia”

“This work is supported by Monash University Malaysia through an internal matching

grant, in collaboration with the Public Service Department (JPA) of Malaysia .”

“Internal Matching Grant - Public Service Department (JPA) of Malaysia”

6. In the online submission form, you indicated that [Data available on request from the author through email.].

Reviewers' comments:

Reviewer's Responses to Questions

**Comments to the Author**

1. Is the manuscript technically sound, and do the data support the conclusions?

Reviewer #1: Partly

Reviewer #2: Yes

2. Has the statistical analysis been performed appropriately and rigorously? 

Reviewer #1: No

Reviewer #2: Yes

3. Have the authors made all data underlying the findings in their manuscript fully available?

Reviewer #1: No

Reviewer #2: No

4. Is the manuscript presented in an intelligible fashion and written in standard English?

Reviewer #1: Yes

Reviewer #2: Yes

5. Review Comments to the Author

Reviewer #1: The paper explores the implementation of gossip protocols in cloud native architectures. Although the topic can be relevant for some use cases, the paper still needs major reworking before being publishable. Below some comments are provided, aiming at being constructive.

The figures are not visible in the manuscript, this gives the impression that the submission is a draft. Moreover, this makes it is very hard to verify the results.

The insights and key findings reported in the conclusion seem trivial and could be easily expected.

Considering the additional network traffic that one could expect when employing gossip protocols, there are serious doubts on the feasibility of using gossip protocols in real-world cloud deployments. This is mostly because currently one of the biggest cost factors is due to the network traffic between availability zones and regions (needed for fault tolerance and high availability), especially in highly distributed cloud native scenarios using microservices architecture. Performance/scalability is relevant, but efficiency can be nowadays even more important. In my opinion, the related work section should focus in relating gossip protocols to other communication patterns suitable to cloud native architectures, instead of comparing to other simulators.

Limited contributions. It seems hard to see how the claim on contribution 1 (Development of an Adaptive Simulator) that such a tool would provide on “translating theoretical concepts of gossip protocols into practical, actionable insights”. Simulators are valid tools, but we are used to expecting the threats to validity about their practical implications. Moreover, it seems difficult to the claimed contribution 2 (a tutorial) as a scientific contribution.

Considering that the research had access to a reasonable setup (Table 4) from Google Kubernetes Engine with 56 CPUs, 96 GB of memory etc, I wonder why the gossip protocol and the deployment still had to be simulated. How far are we from having actual implementations and real-world experiments evaluating the gossip protocols feasibility? Currently, using the abstractions and automations from cloud native architectures should make it possible in a productive way to have realistic test scenarios. In the past, we used simulations mostly because the access to suitable machines was complicated.

Reviewer #2: The manuscript presents a technically sound study exploring the implementation of gossip protocols within cloud-native frameworks. The experiments are rigorously conducted using Google Kubernetes Engine (GKE), and the methodology is detailed, including data creation, collection, and monitoring. The results are clearly presented using histograms and graphs, effectively supporting the conclusions about scalability, resilience, and optimization opportunities for gossip protocols. However, the scalability test is limited to 400 nodes, which may not fully represent large-scale cloud-native environments. Expanding this scope would enhance the validity of the conclusions.

The statistical analysis is appropriate, with results presented in a clear and interpretable manner. Propagation times and other performance metrics are analyzed across different node configurations and network setups, providing meaningful insights into the behavior of gossip protocols. The use of distribution histograms and line graphs effectively illustrates the patterns observed. However, incorporating additional metrics (e.g., CPU usage, network latency) could further enrich the analysis.

The data availability statement indicates that data is available upon request through the author's email. Making the data fully accessible would enhance the study's credibility and allow independent validation of the findings.

The manuscript is well-organized and written in standard English. Technical terms are clearly defined, and the language is precise and accessible to both specialists and non-specialists in cloud computing and network protocols. The logical flow of the introduction, methodology, results, and discussion sections enhances the overall readability.

Recommendations for Improvement:

Scalability Test: Expand the node limit beyond 400 to better represent large-scale cloud-native systems.

Data Availability: Make the dataset publicly accessible through a repository to ensure transparency and reproducibility.

Performance Metrics: Consider adding more metrics such as CPU usage, network latency, and memory consumption for a more holistic performance analysis.

The paper is well-written, and with minor revisions, particularly in scalability testing and data sharing, it would be suitable for publication.

6. PLOS authors have the option to publish the peer review history of their article (what does this mean?). If published, this will include your full peer review and any attached files.

Reviewer #1: No

Reviewer #2: No

---

## [Author Response · Author response to Decision Letter 1]

7 Apr 2025

Response to Editor & Reviewers

We thank the editor and reviewers for their constructive feedback. We have addressed all concerns as follows:

Code & Data Availability: All author-generated code is now publicly available in GitHub (https://github.com/wwiras/cnsim.git), with open-source licensing and documentation. Underlying data has been deposited in [repository name/link] for full reproducibility.

Funder Compliance: We confirm the Public Service Department (JPA) of Malaysia and Monash University provided funding but had no role in study design, execution, or publication decisions. This statement has been added to the cover letter and manuscript.

Key Revisions:

Expanded scalability tests beyond 400 nodes (Reviewer #2)

Added CPU/network metrics for holistic analysis (Reviewer #2)

Restructured the Related Work section to focus on cloud-native communication patterns (Reviewer #1)

Included all missing figures and clarified methodological limitations

Journal Requirements:

Removed funding text from Acknowledgments

Verified PLOS ONE style compliance

Added ethics statement for data collection

Detailed point-by-point responses are provided in the attached "Response to Reviewers" document. Tracked and clean manuscript versions are included.

---

## [Decision Letter · Decision Letter 1]

21 May 2025

Cloud-Native Simulation Framework for Gossip Protocol: Modeling and Analyzing Network Dynamics

PONE-D-24-43101R1

Dear Dr. Samsuddin,

We’re pleased to inform you that your manuscript has been judged scientifically suitable for publication and will be formally accepted for publication once it meets all outstanding technical requirements.

Kind regards,

Jacopo Soldani

Academic Editor

PLOS ONE

Additional Editor Comments (optional):

Reviewers' comments:

Reviewer's Responses to Questions

**Comments to the Author**

1. If the authors have adequately addressed your comments raised in a previous round of review and you feel that this manuscript is now acceptable for publication, you may indicate that here to bypass the “Comments to the Author” section, enter your conflict of interest statement in the “Confidential to Editor” section, and submit your "Accept" recommendation.

Reviewer #1: All comments have been addressed

Reviewer #2: All comments have been addressed

2. Is the manuscript technically sound, and do the data support the conclusions?

Reviewer #1: Partly

Reviewer #2: Yes

3. Has the statistical analysis been performed appropriately and rigorously? 

Reviewer #1: No

Reviewer #2: Yes

4. Have the authors made all data underlying the findings in their manuscript fully available?

Reviewer #1: No

Reviewer #2: No

5. Is the manuscript presented in an intelligible fashion and written in standard English?

Reviewer #1: Yes

Reviewer #2: Yes

6. Review Comments to the Author

Reviewer #1: Comments made in the first revision were somewhat addressed, the authors can use those comments as insights for more practical and advanced research.

Reviewer #2: The authors have addressed the concern I expressed during the previous round of review. Thus, I suggest to accept the manuscript.

7. PLOS authors have the option to publish the peer review history of their article (what does this mean?). If published, this will include your full peer review and any attached files.

Reviewer #1: No

Reviewer #2: No

---

## [Editor Report · Acceptance letter]

PONE-D-24-43101R1

PLOS ONE

Dear Dr. Samsuddin,

I'm pleased to inform you that your manuscript has been deemed suitable for publication in PLOS ONE. Congratulations! Your manuscript is now being handed over to our production team.

Kind regards,

on behalf of

Dr. Jacopo Soldani

Academic Editor

PLOS ONE